# SPARSE AND STRUCTURED VISUAL ATTENTION

## ABSTRACT

Visual attention mechanisms have been widely used in image captioning models. In this paper, to better link the image structure with the generated text, we replace the traditional softmax attention mechanism by two alternative sparsity-promoting transformations: sparsemax and Total-Variation Sparse Attention (TVMAX). With sparsemax, we obtain sparse attention weights, selecting relevant features. In order to promote sparsity and encourage fusing of the related adjacent spatial locations, we propose TVMAX. By selecting relevant groups of features, the TV-MAX transformation improves interpretability. We present results in the Microsoft COCO and Flickr30k datasets, obtaining gains in comparison to softmax. TV-MAX outperforms the other compared attention mechanisms in terms of human-rated caption quality and attention relevance.

## 1 INTRODUCTION

The goal of **image captioning** is to generate a fluent textual caption that describes a given image (Farhadi et al., 2010; Kulkarni et al., 2011; Vinyals et al., 2015; Xu et al., 2015). Image captioning is a multimodal task: it combines text generation with the detection and identification of objects in the image, along with their relations. While neural encoder-decoder models have achieved impressive performance in many text generation tasks (Bahdanau et al., 2015; Vaswani et al., 2017; Chorowski et al., 2015; Chopra et al., 2016), it is appealing to design image captioning models where structural bias can be injected to improve their **adequacy** (preservation of the image's information), therefore strengthening the link between their language and vision components.

State-of-the-art approaches for image captioning (Liu et al., 2018a;b; Anderson et al., 2018; Lu et al., 2018) are based on encoder-decoders with visual attention. These models pay attention either to the features generated by convolutional neural networks (CNNs) pretrained on image recognition datasets, or to detected bounding boxes. In this paper, we focus on the former category: **visual attention** over features generated by a CNN. Without explicit object detection, it is up to the attention mechanism to identify relevant image regions, in an unsupervised manner.

A key component of attention mechanisms is the transformation that maps scores into probabilities, with softmax being the standard choice (Bahdanau et al., 2015). However, softmax is **strictly dense**, i.e., it devotes some attention probability mass to *every* region of the image. Not only is this wasteful, it also leads to "lack of focus": for complex images with many objects, this may lead to vague captions with substantial repetitions. Figure 1 presents an example in which this is visible: in the caption generated using softmax (top), the model attends to the whole image at every time step, leading to a repetition of "bowl of fruit." This undesirable behaviour is eliminated by using our alternative solutions: sparsemax (middle) and the newly proposed TVMAX (bottom).

In this work, we introduce novel visual attention mechanisms by endowing them with a new capability: that of **selecting only the relevant features of the image**. To this end, we first propose replacing softmax with **sparsemax** (Martins & Astudillo, 2016). While sparsemax has been previously used in NLP for attention mechanisms *over words*, it has never been applied to computer vision to attend over *image regions*. With sparsemax, the attention weights obtained are sparse, leading to the selection (non-zero attention) of only a few relevant features. Second, to further encourage the weights of related adjacent spatial locations to be the same (e.g., parts of an object), we introduce a new attention mechanism: **Total-Variation Sparse Attention** (which we dub TVMAX), inspired by prior work in **structured sparsity** (Tibshirani et al., 2005; Bach et al., 2012). With TVMAX, sparsity is allied to the ability of selecting *compact* regions. According to our human evaluation experiments,

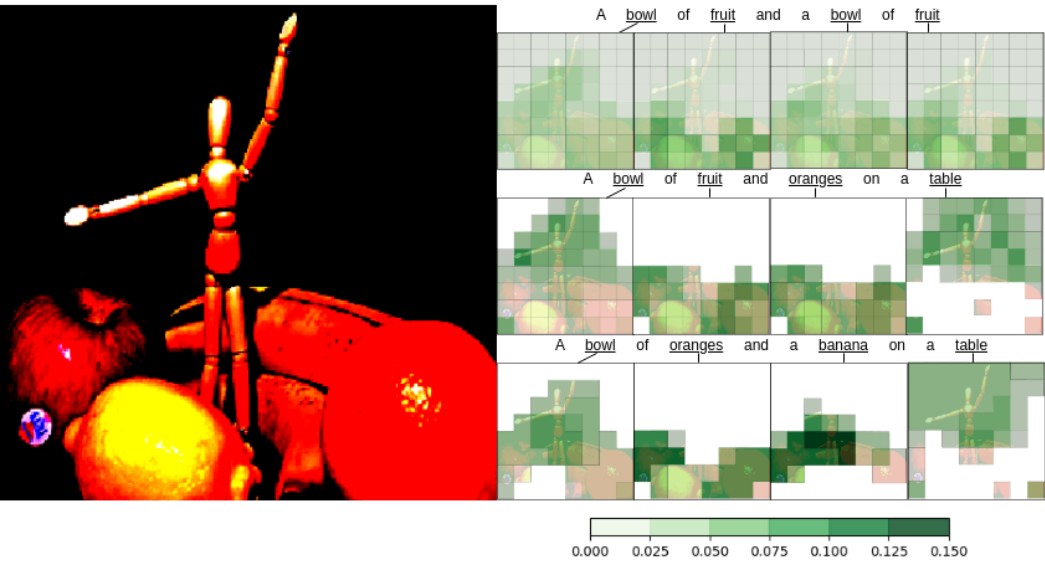

Figure 1: Example of captions generated using softmax (top), sparsemax (middle) and TVMAX attention (bottom). Shading denotes the attention weight, with white for zero attention. The darker the green is, the higher the attention weight is. The full sequences are presented in Appendix C.

this leads to better **interpretability**, since the model's behaviour is better understood by looking at the selected image regions when a particular word is generated. It also leads to a better selection of the relevant features, and consequently to the improvement of the generated captions.

This paper introduces three main contributions:

- We propose a novel visual attention mechanism using sparse attention, based on sparsemax (Martins & Astudillo, 2016), that improves the quality of the generated captions and increases interpretability.

- We introduce a new attention mechanism, TVMAX, that encourages sparse attention over contiguous 2D regions, giving the model the capability of selecting compact objects. We show that TVmax can be evaluated by composing a proximal operator with a sparsemax projection, and we provide a closed-form expression for its Jacobian. This leads to an efficient implementation of its forward and backward pass.

- We perform an empirical and qualitative comparison of the various attention mechanisms considered. We also carry out a human evaluation experiment, taking into account the generated captions as well as the perceived relevance of the selected regions.

## 2 SELECTIVE VISUAL ATTENTION

Attention mechanisms have the ability to select the relevant features, in this case spatial locations. This requires a mapping from importance scores to a distribution, $\boldsymbol{z} \in \mathbb{R}^k \mapsto \boldsymbol{p} \in \triangle^k$, where $\triangle^k := \left\{ \boldsymbol{p} \in \mathbb{R}^k \mid \sum_{i=1}^k p_i = 1, \boldsymbol{p} \geqslant \boldsymbol{0} \right\}$ denotes the simplex (the set of all probability distributions over $k$ values). The standard choice for this mapping is softmax, defined as:

$$[\mathsf{softmax}(\boldsymbol{z})]_i = \frac{\exp(z_i)}{\sum_j \exp(z_j)}. \tag{1}$$

However, as softmax is strictly positive, its output is dense. Thus, the model must pay some attention to the whole image and, consequently, assign lower attention weights to the relevant regions. This motivates our proposed **selective** visual attention mechanisms, which, by being sparse, are able to better isolate the relevant image regions.

## 2.1 SPARSEMAX

To achieve selective capabilities, we propose the use of **sparsemax** (Martins & Astudillo, 2016), a sparse mapping consisting in the Euclidean projection of $\boldsymbol{z}$ onto the probability simplex:

$$\mathsf{sparsemax}(\boldsymbol{z}) := \underset{\boldsymbol{p} \in \triangle^k}{\arg\min} \frac{1}{2}\|\boldsymbol{p} - \boldsymbol{z}\|_2^2, \tag{2}$$

which allows to obtain sparse outputs with a small increase in complexity. Output sparsity is an attractive property for attention mechanisms, since some features do not provide relevant information for the current prediction. In the image captioning case, using sparsemax allows focusing only on the spatial locations of the image that are relevant to the word being generated, assigning zero attention weight to all other regions.

## 2.2 SPARSE AND STRUCTURED VISUAL ATTENTION

To generate descriptive captions, the model should identify the objects present in the image. Thus, when generating object-related words, the attention mechanism should assign high weights to the regions of the image containing the object. However, sparsemax is unstructured and index-invariant, leading it to select discontinuous regions. To overcome this, we propose a new visual attention mechanism, **TVMAX**. TVMAX is a non-trivial generalization of fusedmax (Niculae & Blondel, 2017), a transformation based on fused lasso, to the 2D case. To this end, we first extend fusedmax even more generally, to arbitrary graphs.

### 2.2.1 GENERALIZED FUSED LASSO

Let $\boldsymbol{w} \in \mathbb{R}^k$, and let $I = \{1, \ldots, k\}$. Consider a graph over $I$ defined by its edges $E \subseteq I \times I$, where an edge between $i$ and $j$ means we want to encourage $w_i$ to be close to $w_j$. For simplicity we use $i \sim j$ as shorthand for $(i, j) \in E$.

The generalized fused lasso penalty (Tibshirani et al., 2005) is defined as:

$$\Omega_E(\boldsymbol{w}) = \sum_{i \sim j} |w_i - w_j|. \tag{3}$$

Minimizing $\Omega_E$ encourages "fused" solutions, i.e., it encourages $w_i = w_j$ for $i \sim j$. In particular, its proximal operator[1] can be seen as a **fused signal approximator**, seeking a vector $\boldsymbol{w}$ that approximates $\boldsymbol{z}$ well (in terms of Euclidean distance) and that is encouraged to be fused:

$$\mathsf{prox}_{\lambda\Omega_E}(\boldsymbol{z}) = \underset{\boldsymbol{w} \in \mathbb{R}^d}{\arg\min} \frac{1}{2}\|\boldsymbol{w} - \boldsymbol{z}\|^2 + \lambda\Omega_E(\boldsymbol{w}). \tag{4}$$

**Computing the value** of $\mathsf{prox}_{\lambda\Omega_E}$ is non-trivial in general (Xin et al., 2016), but for certain edge configurations, described below, efficient algorithms exist.

- If $E$ forms a chain, i.e. $i \sim j \iff i = j - 1$, the problem is called **1D total variation** and can be solved in $\mathcal{O}(k)$ time using the *taut string algorithm* (Davies & Kovac, 2001; Barbero & Sra, 2014). We use the quasilinear algorithm of Condat (2013), which is very fast in practice.

- If the indices are aligned on a 2D grid, as in an image, and $i \sim j$ holds iff. $j$ is to the right *or* immediately below $i$, the problem is called **2D total variation**. Unlike the 1D case, exact algorithms are not available. However, for an input of size $a \times b$, it is possible to *split* the penalty into $a$ column-wise and $b$ row-wise 1D problems. We may then apply a number of iterative methods, for instance *proximal Dykstra* (Barbero & Sra, 2014).[2]

---

[1]The proximal operator is defined in Eq. 11 of Appendix A.1.

[2]We use the implementation readily available in the `copt` library, available at `http://openopt.github.io/copt/`.

### 2.2.2 TVMAX

TVMAX combines 2D total variation (TV2D) regularization with sparsemax. This way it promotes sparsity and encourages the attention weights of adjacent spatial locations to be the same, selecting contiguous regions of the image. TVMAX is defined as follows:

**Definition 1** (TVMAX). *Let $z \in \mathbb{R}^k$, such that $z$'s indices can be decomposed into rows and columns. The TVMAX transformation is defined as*

$$\text{TVMAX}(z) := \underset{p \in \triangle^k}{\arg\min} \frac{1}{2} \|p - z\|_2^2 + \lambda \Omega_{2D}^{TV}(p), \tag{5}$$

*where $\lambda$ is an hyper-parameter controlling the amount of fusion ($\lambda = 0$ recovers sparsemax) and $\Omega_{2D}^{TV}$ is a 2D total variation penalty.*

Note that Eq. 5 differs from Eq. 4 in which the variable $p$ is further constrained to lie in the probability simplex. We show next how the forward and backward passes can be efficiently computed.

### 2.2.3 GENERALIZED FUSED SPARSE ATTENTION

To construct **generalized fused sparse attention**, we follow Niculae & Blondel (2017) and define

$$\text{gfusedmax}_E(z) := \underset{p \in \triangle}{\arg\min} \|p - z\|_2^2 + \lambda \Omega_E(p). \tag{6}$$

This can be seen as a *constrained* fused lasso approximator, because the solution $p$ must be a probability distribution vector. While the optimization function is very similar to Eq. 4, the additional constraint that $p \in \triangle$ increases complexity. Fortunately, the following result holds:

**Proposition 1** (Computing generalized fusedmax).

$$\text{gfusedmax}_E(z) = \text{proj}_\triangle \left( \text{prox}_{\lambda \Omega_E}(z) \right). \tag{7}$$

The proof is given in Appendix A.2.

Proposition 1 also provides a shortcut for deriving the Jacobian of generalized fusedmax via the *chain rule*: denoting by $J_F$ the Jacobian of $\text{prox}_{\lambda \Omega_E}$, we have

$$\frac{\partial \, \text{gfusedmax}}{\partial z} = J_{\text{gfusedmax}} = J_{\text{sparsemax}}(\text{prox}_{\lambda \Omega_E}(z)) J_F(z).$$

As we already know how to compute $J_{\text{sparsemax}}$ (Appendix A.1), we may concentrate our effort on deriving the simpler $J_F$ (Eq. 9).

**Proposition 2** (Group-wise characterization of $\text{prox}_{\lambda \Omega_E}$). *Let $w^\star := \text{prox}_{\lambda \Omega_E}$, and denote by $G_i$ the set of indices fused to $w_i$ in the solution, $G_i$ may be defined recursively:*

  *1. $i \in G_i$ for all i, and*

  *2. $j \in G_i$ if there exists $m \in G_i$ such that $m \sim j$ and $w_m^\star = w_j^\star$.*

*Define $s_{ij} = \text{sign}(w_i^\star - w_j^\star)$. Then, the solution has the expression*

$$w_i^\star = \frac{1}{|G_i|} \sum_{j \in G_i} \left( z_j + \sum_{\substack{m \sim j \\ m \notin G_i}} \lambda s_{mj} - \sum_{\substack{j \sim m \\ m \notin G_i}} \lambda s_{jm} \right). \tag{8}$$

Proposition 2 shows how to easily compute a generalized Jacobian of gfusedmax: since small perturbations in $z$ never change the groups $G_i$ nor the signs of across-group differences $s_{ij}$, differentiating Eq. 8 yields

$$J_{F i,j} = \frac{\partial w_i^\star}{\partial z_j} = \begin{cases} \frac{1}{|G_i|}, & j \in G_i, \\ 0, & j \notin G_i. \end{cases} \tag{9}$$

This generalizes Lemma 1 of Niculae & Blondel (2017) to generalized fused lasso, with a simpler proof, given in Appendix A.3.

### 2.2.4 COMPUTATION

As we show in Proposition 1, computing TVMAX's forward pass can be done by chaining efficient algorithms for TV2D and sparsemax.

From Eq. 7 we have that TVMAX's Jacobian can be computed as $\boldsymbol{J}_{\text{TVMAX}} = \boldsymbol{J}_{\text{sp}}(\text{prox}_{\lambda\Omega_{2D}^{TV}}(\boldsymbol{z}))\boldsymbol{J}_{\text{tv}}(\boldsymbol{z})$, where $\boldsymbol{J}_{\text{sp}}$ is the sparsemax's Jacobian and $\boldsymbol{J}_{\text{tv}}$ is the Jacobian of the Total Variation proximal operator.[3] As derived in Proposition 2, $(\boldsymbol{J}_{\text{tv}})_{i,j} = 1/n_{ij}$ if $i$ and $j$ are fused in a group with $n_{ij}$ elements, and 0 otherwise.

The backward pass intuitively involves "spreading" the credit assigned to one image location evenly across all locations fused with it. This can be implemented by Algorithm 1 in $\mathcal{O}(k + N_g \log k)$ where $N_g$ is the number of groups of fused positions. In the worst case, when there are no positions fused, the complexity is $\mathcal{O}(k + k \log k)$. This algorithm is inspired by flood filling algorithms (Burtsev & Kuzmin, 1993).

---

**Algorithm 1** TVMAX backward pass (Jacobian-vector products)

---

1  **Input:** $\boldsymbol{p} = \text{TVMAX}(\boldsymbol{z})$, $\text{d}\boldsymbol{p} \in \mathbb{R}^k$.
2  **Output:** $\text{d}\boldsymbol{z} = \boldsymbol{J}_{\text{TVMAX}}^{\top}(\text{d}\boldsymbol{p}) \in \mathbb{R}^k$          *# chain rule*
3  **Initialize:** $N \leftarrow \varnothing$          *# neighbours stack*
4          $V \leftarrow \varnothing$          *# visited positions*
5          $G \leftarrow \varnothing$          *# current group*
6          $s = 0$          *# intermediate value used for $J_{\text{TVMAX}}$'s computation*
7  $\text{d}\boldsymbol{w} \leftarrow (\boldsymbol{J}^{\text{sp}})^{\top} \text{d}\boldsymbol{p}$          *# Eqs. 14 and 15 of §A.1*
8  **while** $|V| < k$ **do**          *# check if all positions have been visited*
9      **pick** $(i_0, j_0) \notin V$, **push** $(i_0, j_0)$ **to** $N$          *# get not visited position and add it to neighbours stack*
10     **while** $N$ not empty **do**
11         **pop** $(i, j)$ **from** $N$          *# get element from neighbours stack*
12         **if** $p_{i,j} = p_{i_0,j_0}$ **then**          *# check if element is fused*
13             $G \leftarrow G \cup \{(i,j)\}, V \leftarrow V \cup \{(i,j)\}$          *# add neighbour to group and to visited positions*
14             $s \leftarrow s + (\text{d}\boldsymbol{w})_{i,j}$          *# sum of the $\text{d}\boldsymbol{w}$ of each element of the group*
15             **for all** neighbours $(i', j') \sim (i, j)$ **do**
16                 **if** $(i', j') \notin V$ **then push** $(i', j')$ **to** $N$          *# add not visited neighbours of $(i, j)$ to the stack*
17     **if** $G$ not empty **then:**
18         $(\text{d}\boldsymbol{z})_{i,j} \leftarrow s/|G|$ for all $(i, j) \in G$          *# compute $J_{\text{TVMAX}}$ for elements in group $G$*
19         $G \leftarrow \varnothing$
20         $s = 0$

---

## 3 IMAGE CAPTIONING MODEL

To compare the proposed attention mechanisms, we use a straight-forward simple encoder-decoder model with visual attention, inspired by Liu et al. (2018a). The model is sketched in Figure 2.

Given an image, we use a residual CNN pretrained on ImageNet (He et al., 2016; Russakovsky et al., 2014) to get a feature map with spatial dimension of size $8 \times 8$ and channel dimension of size 2048, that go through a fine-tuned feedforward layer yielding $g = 512$ feature maps. The visual feature matrix $\boldsymbol{V} = [v_1, v_2, \ldots, v_k]$, with $v_i \in \mathbb{R}^g$ and $k = 64 = 8 \times 8$, contains the image information used to generate the corresponding caption. Following Liu et al. (2018a), we use input and output attention to select the relevant features for the current generation. To generate the word at position $t$, the **input attention**, $\boldsymbol{\alpha}_t$, is computed using the LSTM's previous hidden state, $\boldsymbol{h}_{t-1} \in \mathbb{R}^d$. First, a similarity score $z_{t,i}, i \in \{1, \ldots, k\}$, is computed between $\boldsymbol{h}_{t-1}$ and the $i^{th}$ image cell via a feedforward transformation (Bahdanau et al., 2015), as $z_{t,i} = \boldsymbol{w}^{\top}\tanh(\text{affine}([\boldsymbol{v}_i; \boldsymbol{h}_{t-1}]))$, for all $k$ image cells. Then, $\boldsymbol{\alpha}_t$ is obtained by normalizing the $k$-dimensional vector of scores $\boldsymbol{z}_t$ with softmax, $\boldsymbol{\alpha}_t = \text{softmax}(\boldsymbol{z}_t)$. Using these attention weights, a vector representation of the image to be used as input of the LSTM, is obtained, $\boldsymbol{s}_t = \boldsymbol{V}\boldsymbol{\alpha}_t$. The **output attention** $\widetilde{\boldsymbol{\alpha}}_t$, is

---

[3] $\boldsymbol{J}_{\text{tv}}$ is a special case of $\boldsymbol{J}_{\text{F}}$, when using the graph $E$ that consists in a 2D grid.

computed in the same way as above, but applied to the current LSTM hidden state $h_t$, instead of $h_{t-1}$, and normalized with the different proposed transformations. This produces output visual features $\widetilde{s}_t = V\widetilde{\alpha}_t$, which are passed through a feedforward layer to yield the image representation $r_t = \tanh(\text{affine}(\widetilde{s}_t))$. Finally, the predictive probability of the next word is:

$$P(y_t \mid y_{1:(t-1)}; \text{Image}) \propto \text{softmax}(\text{affine}([r_t; h_t])). \qquad (10)$$

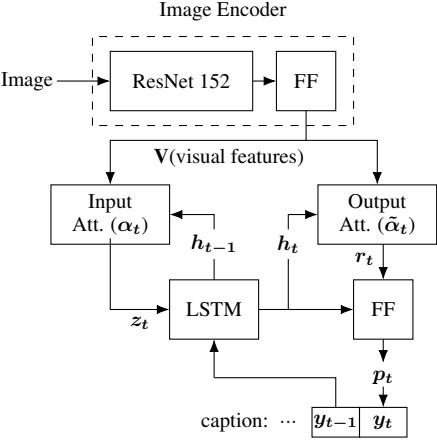

Figure 2: Diagram of the caption generation network.

## 4    EXPERIMENTS

**Settings.**    The input images are resized to $256 \times 256$ before going through the residual CNN and the feature maps obtained have a size of $8 \times 8$. We use an LSTM hidden size of $d = 512$ and a word embedding size of $256$, for all models. The models were trained for $50$ epochs using the Adam optimizer (Kingma & Ba, 2014) with a learning rate of $0.0001$ and a decay of $0.8$ and $0.999$ for the first and second momentum, respectively. After the $10^{th}$ epoch, the learning rate starts decaying with a decay factor of $0.99$. For TVMAX, we set $\lambda = 0.01$.

**Datasets and Metrics.**    We report our results on the Microsoft COCO (MSCOCO) and Flickr30k datasets. MSCOCO is composed of 113,287 images of common objects in context while Flickr30k consists in 31,000 pictures of people involved in everyday activities and events. Each image is annotated with 5 captions. We use the split proposed by Karpathy & Fei-Fei (2015), which stipulates equal validation and test sizes of 5,000 images (MSCOCO) and 1,000 (Flickr30k). The metrics we report are SPICE (Anderson et al., 2016), CIDEr (Vedantam et al., 2015), longest common subsequence ROUGE, (denoted $\text{ROUGE}_L$; Lin, 2004), 1– to 4–gram BLEU (denoted $\text{BLEU}_4$; Papineni et al., 2002), and METEOR (Banerjee & Lavie, 2005). To investigate whether selective attention alleviates repetition, we also measure the n-gram repetition metric REP (Malaviya et al., 2018).

Table 1: Automatic evaluation of caption generation on MSCOCO and Flickr30k.

|  | MSCOCO | | | | | | Flickr30k | | | | | |
|---|---|---|---|---|---|---|---|---|---|---|---|---|
|  | SPICE | CIDER | $\text{ROUGE}_L$ | $\text{BLEU}_4$ | METEOR | REP↓ | SPICE | CIDER | $\text{ROUGE}_L$ | $\text{BLEU}_4$ | METEOR | REP↓ |
| softmax | 18.4 | 0.967 | 52.9 | 29.9 | 24.9 | 3.76 | 13.5 | 0.443 | 44.2 | 19.9 | 19.1 | 6.09 |
| sparsemax | **18.9** | **0.990** | **53.5** | **31.5** | **25.3** | 3.69 | **13.7** | **0.444** | **44.3** | **20.7** | **19.3** | 5.84 |
| TVMAX | 18.5 | 0.974 | 53.1 | 29.9 | 25.1 | **3.17** | 13.3 | 0.438 | 44.2 | 20.5 | 19.0 | **3.97** |

**Automated metrics.**    As can be seen in table 1, overall sparsemax and TVMAX attention mechanisms achieve better results when compared with softmax, indicating that the use of selective attention leads to better captions. This improvement does not come at a high computational cost: at

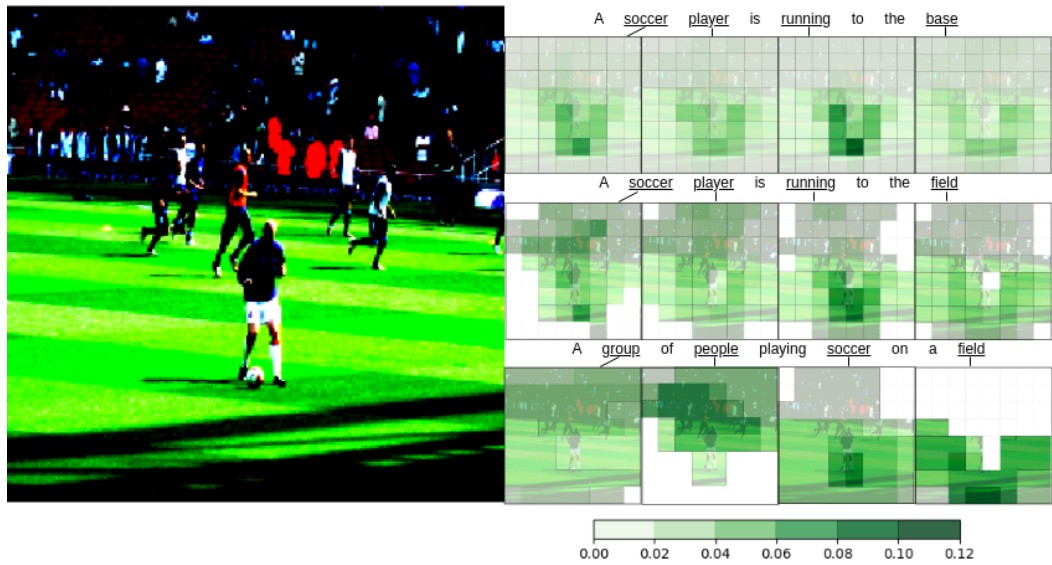

Figure 3: Example of captions generated using softmax (top), sparsemax (middle) and TVMAX attention (bottom). Shading denotes the attention weight, with white for zero attention. The darker the green is, the higher the attention weight is. The full sequences are presented in Appendix C.

inference time, models using TVMAX and sparsemax are only 1.3x and 1.1x slower than softmax. Moreover, for TVMAX, automatic metrics results are slightly worse than sparsemax but still superior to softmax on MSCOCO and similar on Flickr30k. We show next that this is compensated with fewer repetitions and higher scores in the human evaluation of the captions and attention relevance.

Table 2: Human evaluation results with different attention mechanisms on MSCOCO.

|  | CAPTION (1-5) | ATTENTION RELEVANCE (1-5) |
| --- | --- | --- |
| softmax | 3.50 | 3.38 |
| sparsemax | 3.71 | 3.89 |
| TVMAX | **3.87** | **4.10** |

**Human rating.** The caption evaluation consisted in attributing a score from 1 to 5 to the caption of each model while the attention evaluation consisted in scoring the relevancy of the attended areas, from 1 to 5, when generating the non stop words of the captions. A full description of the human assessment can be found in Appendix B.

Despite performing slightly worse than sparsemax under automated metrics, TVMAX outperforms sparsemax and softmax in the caption human evaluation and the attention relevance human evaluation, reported in Table 2. The superior score on attention relevance shows that TVMAX is better at selecting the relevant features and its output is more interpretable. Additionally, the better caption evaluation results demonstrate that the ability to select compact regions induces the generation of better captions. We next explore possible explanations for the TVMAX superior results.

**Repetition.** Figure 1 illustrates that softmax attention is prone to spuriously repeating references to the same object. Selective attention mechanisms like sparsemax and especially TVMAX reduce repetition, as measured by the REP metric reported in Table 1. This expected success can be attributed to the sparsity of the attention weights distribution and to the ability to select compact regions exclusively and can be one of the causes of the human evaluation results. This happens even though TVMAX generates longer sentences than sparsemax and softmax (9.5 against 9.0 words on average) and shows the benefit of promoting structured and sparse attention simultaneously. To corroborate

our intuition that sparsity leads to less repetition, we measured the Jensen-Shannon divergence (JS) between the attention distributions for each step of the generation of the captions correspondent to the images of the MSCOCO test set. The mean JS values are 0.12, 0.29, and 0.34 for softmax, sparsemax, and TVmax, respectively. This shows that sparsity leads to less similar attention distributions along the generation of the captions and, consequently, to less repetitions.

**Object detection.** Using the MSCOCO object detection ground truth, we compared the percentage of objects present in the image that are referred to in the captions, using each attention mechanism. With TVMAX 28.2% of the reference objects are referred, against 27.5% and 27.4% for sparsemax and softmax, repectively. This shows that promoting high attention to groups of spatial locations of the image leads to a more precise identification of the objects.

**Sparsity.** The average image area that receives zero attention is 34% for sparsemax and 25% for TVMAX. To illustrate where the models attend to, we display the output attention in Figures 1 and 3. As expected, softmax weights are spread widely across the image, ending up missing the relevant regions. In contrast, sparsemax and TVMAX weights are zero for the non-relevant spatial locations.

**Qualitative comparison.** As the image of Figure 1 contains various similar objects, the softmax model (top) generates a incoherent, repetition-laden caption. In contrast, the sparsemax (middle) and TVMAX (bottom) models better identify the relevant parts of the image, generating coherent and descriptive captions. Moreover, the groups obtained with TVMAX are clearly visible and more aligned to object boundaries, offering better interpretability, as revealed by human attention assessment. In Figure 3 it can also be noticed that with TVMAX (bottom) the model correctly identified "a group of people" instead of "a soccer player" as with sparsemax (middle) and softmax (top). This indicates its superior ability to correctly define the relevant groups of features and that this ability leads to improved captions.

## 5 RELATED WORK

**Image captioning.** In the last years, neural models with visual attention mechanisms have been receiving increased interest. Several researchers have been studying diverse attention mechanisms in order to refine visual information for image captioning. Xu et al. (2015) proposed the use of hard attention, which only attends to one region at each step. However, to generate descriptive captions the model should, often, focus on more than one region. In addition, hard attention is non-differentiable, requiring imitation learning or Monte Carlo policy gradient approximations.Anderson et al. (2018) proposed bottom-up attention, using an object detection model designed to identify bounding boxes of objects, and top-down attention, selecting the relevant bounding-boxes. Wang et al. (2019) proposed an hierarchical attention network composed by a patch detector, object detector, and concept detector. Using object detection models is less demanding on the attention mechanism, since it only has to select the boxes the model should attend to. However, such models are limited by the bounding boxes position's accuracy. Gao et al. (2019) introduced a deliberate attention network to refine the attended visual features. Yet, the attention distribution remained dense.

**Sparse attention.** In several tasks only a few features are relevant for the current prediction. This can be attained when using sparse attention. Various prior works have proposed sparse attention mechanisms with promising results, (Xu et al., 2015; Martins & Astudillo, 2016; Malaviya et al., 2018; Peters et al., 2019). Niculae & Blondel (2017) proposed 1D fusedmax, which incorporates the fused lasso, so that adjacent words are encouraged to have the same attention weight. In this work, the authors were able to improve interpretability without sacrificing performance, obtaining superior results on textual entailment and summarization. We derive a generalized fused attention mechanism, extending 1D fusedmax.

## 6 CONCLUSIONS AND FUTURE WORK

We propose using sparse and structured visual attention, in order to improve the process of selecting the features relevant to the caption generation. For that, we used sparsemax and introduced TVMAX. Results on the image captioning task, show that the attention mechanism is able to select better

features when using sparsemax or TVMAX. Furthermore, in the human assessment and attention analysis we see that the improved selection of the relevant features as well as the ability to group spatial features lead to the generation of better captions, while improving the model's interpretability.

In future work, TVMAX attention can be applied to other multimodal problems such as visual question answering. It can also be applied in other tasks for which we have prior knowledge of the data's stucture, for instance graphs or trees.

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

## A FORWARD AND BACKWARD PASS OF 2D FUSEDMAX ATTENTION.

### A.1 PRELIMINARIES

The **proximal operator** of a function $f \colon \mathbb{R}^d \to \mathbb{R} \cup \{\infty\}$ is defined as

$$\mathsf{prox}_f(\boldsymbol{z}) = \underset{\boldsymbol{w} \in \mathbb{R}^d}{\arg\min}\, f(\boldsymbol{z}) + \frac{1}{2}\|\boldsymbol{z} - \boldsymbol{w}\|_2^2, \tag{11}$$

and it is guaranteed to have a unique solution, thanks to the strong convexity of the Euclidean distance.

The **indicator function** of a set $\mathcal{C} \subset \mathbb{R}^d$ is the function

$$\iota_{\mathcal{C}} \colon \mathbb{R}^d \to \mathbb{R} \cup \{\infty\}, \quad \iota_{\mathcal{C}}(\boldsymbol{w}) := \begin{cases} 0, & \boldsymbol{w} \in \mathcal{C}, \\ \infty, & \boldsymbol{w} \notin \mathcal{C}. \end{cases} \tag{12}$$

The **projection onto a convex set** $\mathcal{C} \subset \mathbb{R}^d$ is defined as

$$\mathsf{proj}_{\mathcal{C}}(\boldsymbol{z}) := \underset{\boldsymbol{w} \in \mathcal{C}}{\arg\min}\, \frac{1}{2}\|\boldsymbol{z} - \boldsymbol{w}\|_2^2 = \mathsf{prox}_{\iota_{\mathcal{C}}}(\boldsymbol{z}), \tag{13}$$

showing that the proximal operator can be seen as a generalization of projection.

The **sparsemax** attention mapping (Martins & Astudillo, 2016) is the projection onto the simplex,

$$\mathsf{sparsemax}(\boldsymbol{z}) := \mathsf{proj}_\triangle(\boldsymbol{z}) = \underset{\boldsymbol{p} \in \triangle}{\arg\min}\, \frac{1}{2}\|\boldsymbol{p} - \boldsymbol{z}\|^2. \tag{14}$$

A necessary component for using sparsemax for attention is its Jacobian, the matrix of its partial derivatives $(\boldsymbol{J}_{\mathsf{sparsemax}})_{i,j} = \frac{\partial\, \mathsf{sparsemax}(\boldsymbol{z})_i}{\partial z_j}$. Martins & Astudillo (2016) derive its expression

$$\boldsymbol{J}_{\mathsf{sparsemax}}(\boldsymbol{z}) = \mathsf{diag}\, \boldsymbol{s} - \frac{1}{\|\boldsymbol{s}\|_1}\boldsymbol{s}\boldsymbol{s}^\top, \tag{15}$$

where $s_j = 1$ if $\mathsf{sparsemax}(z)_j > 0$ and $s_j = 0$ otherwise.

### A.2 PROOF OF PROPOSITION 1

*Proof.* This result is a slight extension of Proposition 2 in Niculae & Blondel (2017), and also follows from Corrolary 4 of Yu (2013), by taking $f = \iota_\triangle$, and noting that $\iota_\triangle$ is symmetric: if $\boldsymbol{p} \in \triangle$, then any vector $\boldsymbol{p}'$ obtained by permuting $\boldsymbol{p}$ is also in $\triangle$, because its values remain non-negative and sum to 1. □

### A.3 PROOF OF PROPOSITION 2

Let $\boldsymbol{w}^\star := \mathsf{prox}_{\lambda \Omega_E}$, and denote by $G_i$ the set of indices fused to $w_i$ in the solution. Define $s_{ij} = \mathsf{sign}(w_i^\star - w_j^\star)$.

*Proof.* The subgradient optimality conditions of Eq. 4 are: (Friedman et al., 2007)

$$w_i^\star - z_i + \sum_{k:i \sim k} \lambda t_{ik} - \sum_{k:k \sim i} \lambda t_{ki} = 0, \qquad 1 \leq i \leq d. \tag{16}$$

where $t_{ij} = \mathsf{sign}(w_i^\star - w_j^\star)$ if $w_i^\star \neq w_j^\star$, otherwise $t_{ij}$ is a free variable in $[-1, 1]$.

We focus on a single group $G = G_i$, dropping the index $i$ for brevity. Within a fused group, the solution is constant, i.e., $w_j^\star = w$ for $j \in G$. We separate the sums in Eq. 16 according to whether $k \in G$ or not, and move the "constant" terms to the right hand side, yielding the system

$$w + \sum_{\substack{j \sim k \\ k \in G}} \lambda t_{jk} - \sum_{\substack{k \sim j \\ k \in G}} \lambda t_{kj} = z_j + \sum_{\substack{k \sim j \\ k \notin G}} \lambda s_{kj} - \sum_{\substack{j \sim k \\ k \notin G}} \lambda s_{jk}, \qquad j \in G. \tag{17}$$

Summing up the Eq. 17 over all $j \in G$, we observe that for any $k \in G$, the term $\lambda t_{jk}$ appears twice with opposite signs. Thus,

$$\sum_{j \in G} w = \sum_{j \in G} \left( z_j + \sum_{\substack{k \sim j \\ k \notin G}} \lambda s_{kj} - \sum_{\substack{j \sim k \\ k \notin G}} \lambda s_{jk} \right). \tag{18}$$

Dividing by $|G|$ gives exactly Eq. 8. This reasoning applies to any group $G_i$. $\qquad\square$

## B  HUMAN EVALUATION DESCRIPTION

To perform the human evaluation firstly 100 images were randomly selected from the test set of the MSCOCO dataset (using the split proposed by Karpathy & Fei-Fei (2015)). For each of the selected images, the human evaluators selected a score from 1 to 5 for the captions generated by the models using softmax attention, sparsemax attention, and TVMAX attention. They were also asked to evaluate whether the models attend to the relevant regions of the image when generating a certain word. For that they observed the attention plots corresponding to the non stop words of the caption of each of the models. While in Figures 1 and 3, 4, and 5 we emphasized sparsity with a hard white mask, for the human evaluation the sparse regions of the attention plots were simply fully transparent, to avoid biasing the evaluators. The possible scores were also between 1 and 5. The 100 images were judged by 6 persons both for the captions evaluation and attention evaluation. The order of the captions and attention plots was randomly chosen for each image.

With these scores, we computed the mean of the captions evaluation scores and the mean of the attention relevance evaluation scores. The results are reported in Table 2.

## C  ADDITIONAL ATTENTION VISUALIZATION

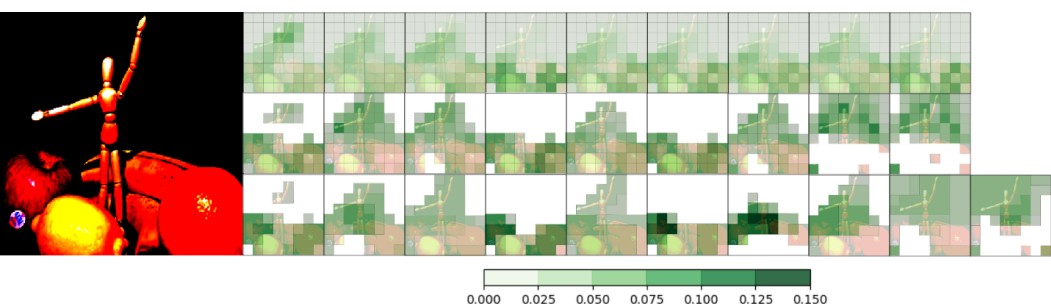

Figure 4: Example generated captions using softmax attention (top), sparsemax attention (middle) and TVMAX attention (bottom). The captions are "A bowl of fruit and a bowl of fruit", "A bowl of fruit and oranges on a table" and "A bowl of oranges and a banana on a table".

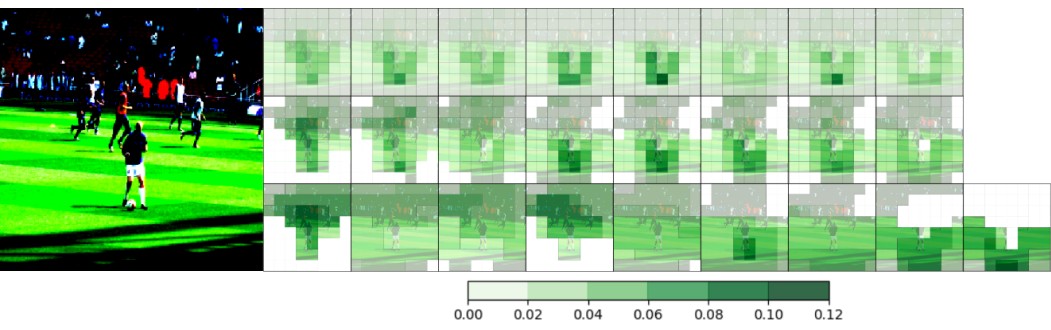

Figure 5: Example generated captions using softmax attention (top), sparsemax attention (middle) and TVMAX attention (bottom). The captions are "A soccer player is running to the base", "A soccer player is running to the field" and "A group of people playing soccer on a field".

