# OpenReview forum: "Sparse and Structured Visual Attention"
_ICLR.cc/2020/Conference — Reject_

### Official Review · AnonReviewer2 · 2019-10-23
**Official Blind Review #2**

**Rating:** 3

**Review:**

This paper studies the problem of applying attention to the problem of image captioning. To this end the authors first apply full attention where the probabilities are computed using softmax before applying the recently proposed Sparsemax - which essentially computes probabilities from scores by performing a projection on to the probability simplex. The authors then propose a variant of Sparsemax which they call TVMax, which has the property that it encourages assigning probability weights to contiguous regions in 2D space unlike Sparsemax which has no such incentive. The main idea is to augment the Sparsemax projection loss with a Lasso like penalty which penalizes assigning different attention probabilities to contiguous regions in the image. The authors then compare their TVMax approach with softmax and Sparsemax attention for image captioning and show improvements on the MSCOCO and Flickr datasets.

The idea of applying additional structural constraints on the sparsity structure induced by Sparsemax is a cool idea, and I like the idea of incentivizing contiguous pixels to have similar attention probabilities. Sparse attention patterns seems like an important direction of research with the motivation of either 1) improving generalization over full attention or 2) scaling to inputs of length where full attention is not feasible. This seems like a good progress in the first direction. The major weakness I see in this work is that the authors only limited their experiments to image captioning. It would be interesting to see if their approach could benefit other tasks such as machine translation, image generation etc. The other issue with their approach is that it doesn't seem to scale well - if I understand correctly their algorithm takes O(n^2logn) for sequence length n. The other issue potentially could be weak baselines since the authors use an LSTM for their caption generation network instead of Transformer.

[Edit: After going through reviewer discussion, I updated my score to reject. I am not convinced of the motivation for sparse attention unless it is for long sequences, since otherwise the regular softmax should be able to assign 0's to the un-needed items. Moreover, for generalization one can use attention dropout which is simpler instead.]

**Experience Assessment:**

I have published one or two papers in this area.

**Review Assessment: Checking Correctness Of Derivations And Theory:**

I assessed the sensibility of the derivations and theory.

**Review Assessment: Checking Correctness Of Experiments:**

I assessed the sensibility of the experiments.

**Review Assessment: Thoroughness In Paper Reading:**

I read the paper at least twice and used my best judgement in assessing the paper.

---

> ### Author Response · Authors · 2019-11-12
> **Author Response to Blind Review #2**
>
> Thank you for your positive comments! We are glad that you think that inducing sparsity and structure in the attention transformation is a good idea.
>
> Actually, TVmax’s backward pass has a complexity of O($n+n \log n$) for the worst case: when there are no positions fused (we had made a mistake in the first version of the paper: the first loop over n corresponds sparsemax’s backward pass, so it should be summed and not multiplied; this has been updated in the new version of the paper).
> As a function of the number of groups of fused positions ($N_g$), the complexity is O($n+N_g \log n$). Thus, in practice it is much faster than in the worst case. Also, at inference time the model using TVmax is only 1.3 times slower than the model using softmax.
>
> Thanks for the suggestion of applying this technique in other tasks. Prior work [1,2,3,4] applied sparsemax successfully to NLP tasks such as machine translation, entailment, and summarization. In this paper we focus on visual attention and on extending fusedmax to 2D, yielding TVmax. We are currently experimenting TVmax in other vision tasks. We believe that our paper may open the door to future application in other tasks.
>
> [1] C. Malaviya, P. Ferreira, and A. Martins. Sparse and constrained attention for neural machine translation. ACL 2018. https://www.aclweb.org/anthology/P18-2059.pdf
>
> [2] A. Martins and R. Astudillo. From softmax to sparsemax: A sparse model of attention and multi-label classification. ICML 2016. http://proceedings.mlr.press/v48/martins16.pdf
>
> [3] V. Niculae and M. Blondel. A regularized framework for sparse and structured neural attention. NeurIPS 2017.
> https://papers.nips.cc/paper/6926-a-regularized-framework-for-sparse-and-structured-neural-attention.pdf
>
> [4] B. Peters, V. Niculae, and A. Martins. Sparse sequence-to-sequence models. ACL 2019.
> https://www.aclweb.org/anthology/P19-1146.pdf

---

### Official Review · AnonReviewer3 · 2019-10-26
**Official Blind Review #3**

**Rating:** 3

**Review:**

This paper proposes two sparsifying methods of computing attention weights, dubbed sparsemax and TVmax, which appear to slightly improve objective and subjective image captioning scores.    The sparsifying projections are posed as optimization problems, and algorithms for their computation, along with formula for their gradients are given.  Proof of the optimality of these algorithms relies significantly on prior work, so could not be checked deeply without bringing in additional sources.

It is not clear that the motivation for these sparsifying objectives is sound.   The conventional softmax approach to attention weights should be capable of producing attention weights near zero, which would be effectively sparse, especially if the pre-activations, z_i, in equation (1), are allowed to have a large enough range.   It's not clear why weights should need to be zero exactly in the ignored regions, since being near zero should be sufficient to contribute almost nothing to the subsequent weighted sum.    So it is also not clear why the strict sparsity itself, as opposed to the effective sparsity of the softmax, should explain the differences in Figure 1, and in the results.  In particular it is unclear why the strict sparsity should prevent repetition; when looking at the weight distributions in the two cases, a more likely story seems to be that the weight distributions don't repeat as much from one word to the next in the second case, but there is no clear reason to attribute this to sparsity.   The pictures of the attention weights are lacking a color scale, so it is impossible to see how close to zero it comes in the unattended regions, although the gray color values chosen for these regions might be misleading.
The TVmax approach, in addition to sparsity, also constrains the non-zero region to be contiguous.   To the extent that this improves performance, this presumably introduces an inductive bias that matches the data.   It is unclear why this fails to produce better objective scores than sparsemax, while producing better human ratings.   In any case it is not clear why this should necessarily be a good inductive bias for all images, although it is plausible that it helps in some cases.
In many neural network problems, what makes a difference has more to do with the optimizability of the gradients, than the specific activations per se, and that might be the case here too, although the paper does not analyze this aspect of the proposed models.

Overall the paper is flawed by the lack of clarity in the motivation for the proposed methods, and the lack of retrospective analysis and understanding of why the proposed methods should improve results.


**Experience Assessment:**

I have published one or two papers in this area.

**Review Assessment: Checking Correctness Of Derivations And Theory:**

I assessed the sensibility of the derivations and theory.

**Review Assessment: Checking Correctness Of Experiments:**

I assessed the sensibility of the experiments.

**Review Assessment: Thoroughness In Paper Reading:**

I read the paper thoroughly.

---

> ### Author Response · Authors · 2019-11-12
> **Author Response to Blind Review #3**
>
> Thank you for your detailed comments. The main points you raised are related to 1) the motivation for our proposed method and 2) the analysis and understanding of why it improves results. We clarify these two points below and we added new analyses to the revised version of the paper to help understanding where the improvements come from.
>
> 1) Motivation for our proposed method.
> While softmax may lead to attention weights near zero, prior work [1,2,3,4] shows that it has the tendency of accumulating too much probability mass on a long tail of irrelevant features, which may harm model performance. In contrast, sparse attention mechanisms are able to select only a small set of features, with improved attention focus. In the refs above, the success of sparse attention has been shown for NLP tasks such as machine translation, textual entailment, summarization, and morphological inflection. The main motivation for this work is to test if this hypothesis also holds for a significantly different task, image captioning, where attention is visual. Our findings confirm those of Xu et al. [5], who obtained better results and improved interpretability with hard attention over softmax. However, their approach is not end-to-end differentiable, requiring imitation learning or Monte Carlo policy gradient approximations, while ours is differentiable and can be used as a drop-in replacement for softmax. To further induce structural bias on top of sparsity (important when we want to exploit spatial correlations in visual tasks), we proposed TVmax, that promotes selection of contiguous regions. We will make these motivations clearer in the final version of the paper.
>
> 2) Analysis and understanding of why it improves results.
> Thank you for suggesting a color scale to facilitate the visualization of the attention plots -- we included it in the new version of the paper. The sparsity benefits are qualitatively illustrated in Figure 1 and 3: we can observe that when using sparsemax and TVmax the weights given to the relevant regions are much higher than the ones given when using softmax, while the non relevant regions receive zero attention. This is corroborated by the human attention evaluation. It also confirms our motivation for TVmax that by focusing in contiguous regions, high attention weights can be attributed to compact objects, improving their detection.
> Moreover, human evaluation scores and automatic REP scores are considerably higher when using TVmax. This happens even though TVmax generates longer sentences.
>
> We posit that the reasons behind the decrease in the number of repetitions are two-fold:
> - By using softmax, some attention (even if small) is given to all regions of the image in every time step, even those that are not relevant for the word being generated (see point 1 above). Consequently, the feature vector obtained is more similar between time steps than with sparsemax or TVmax, possibly leading to the generation of the same word repeatedly. This is in agreement with your intuition that the smaller number of repetitions is caused by the weight distributions not repeating as much.
> - Focusing on compact regions of the image leads to the detection of more objects, which can lead to less repetitions.
>
> To corroborate the first point, we measured the Jensen-Shannon divergence (JS) between the attention probabilities for each time step of the captions correspondent to the MSCOCO test set images. The mean JS values are 0.12, 0.29, and 0.34 for softmax, sparsemax, and TVmax, respectively. This confirms our intuition that sparsemax and TVmax lead to less similar attention distributions across time steps and, consequently, to less repetitions. We report this statistic in the new version of the paper.
>
> Note that this decrease of repetitions is consistent with previous findings in sequence-to-sequence models (machine translation) which has shown that sparsemax has much lower propensity for repetitions than softmax (see ref [1] below, Table 1, which reports consistently better REP scores for several language pairs).
>
> [1] C. Malaviya, P. Ferreira, and A. Martins. Sparse and constrained attention for neural machine translation. ACL 2018. https://www.aclweb.org/anthology/P18-2059.pdf
>
> [2] A. Martins and R. Astudillo. From softmax to sparsemax: A sparse model of attention and multi-label classification. ICML 2016. http://proceedings.mlr.press/v48/martins16.pdf
>
> [3] V. Niculae and M. Blondel. A regularized framework for sparse and structured neural attention. NeurIPS 2017.
> https://papers.nips.cc/paper/6926-a-regularized-framework-for-sparse-and-structured-neural-attention.pdf
>
> [4] B. Peters, V. Niculae, and A. Martins. Sparse sequence-to-sequence models. ACL 2019.
> https://www.aclweb.org/anthology/P19-1146.pdf
>
> [5] K. Xu, J. Ba, R. Kiros, K. Cho, A. Courville, R. Salakhudinov, R. Zemel, and Y. Bengio. Show, attend and tell: Neural image caption generation with visual attention. ICML 2015. http://proceedings.mlr.press/v37/xuc15.pdf

---

### Official Review · AnonReviewer4 · 2019-11-02
**Official Blind Review #376**

**Rating:** 6

**Review:**

This paper produces a new method call TVmax and presents that the selective visual attention could improve the score in the Image Captioning task. Different from the fusedmax[2] which fuses attention in one dimension, the proposed method encourages the sparse attention over contiguous 2D regions. Compared with the softmax function, the sparsemax[1]  and the TVmax are able to sparse the visual attention very well. The paper also evaluates the score in both automated metrics and the human rating. Experiments show that the sparse visual attention achieves higher performance with a little computational cost.

One problem in this paper is that the author applies their proposed TVmax on Image Captioning  task, however it only achieves a little improvement on the automated metrics compared with the baseline(softmax). I wonder whether there is a better task for evaluating the visual attention.

Although the proposed method (TVmax) is slightly worse than the sparsemax in the automated metrics, it is still promising in multimodal problems.

Therefore, My decision leans to a weak accept.

Some questions:
1.From the experiments, the proposed method achieved only a little higher performance than the baseline(softmax). Could you please show some reasons about that?
2.Could you show some results of TVmax on the other task in order to show the effectiveness of the proposed method?


[1]From Softmax to Sparsemax: A Sparse Model of Attention and Multi-Label Classification.      André F. T. Martins, Ramón Fernandez Astudillo
[2]A Regularized Framework for Sparse and Structured Neural Attention .            Vlad Niculae, Mathieu Blondel

**Experience Assessment:**

I have read many papers in this area.

**Review Assessment: Checking Correctness Of Derivations And Theory:**

I assessed the sensibility of the derivations and theory.

**Review Assessment: Checking Correctness Of Experiments:**

I carefully checked the experiments.

**Review Assessment: Thoroughness In Paper Reading:**

I read the paper thoroughly.

---

> ### Author Response · Authors · 2019-11-12
> **Author Response to Blind Review #376**
>
> Thank you for your positive comments!
>
> One reason for the improvement achieved being bigger on the human evaluation than on automated metrics can be related with the fact that most of the commonly used metrics for image captioning are based on n-gram overlap (CIDER, ROUGE, BLEU, METEOR) and do not convey enough information about the quality of the captions. The metric SPICE compares scene graphs, yielding information concerning the recovery of objects, attributes, and the relations between them. However, this is dependent on the quality of the semantic parser. Furthermore, other properties of the caption, such as its coherence, complexity, syntax, and amount of repetitions, are not considered.
> These are the reasons that led us to perform human evaluation. The human results showed us a big improvement when using TVmax and sparsemax over softmax attention. It can also be seen in Table 1 that TVmax leads to a smaller number of repetitions (REP score).
> We also show qualitatively, using examples, that the use of the proposed transformation results in higher attention weights over regions of the image that are relevant to the word being generated. This is corroborated by the human attention evaluation.
> Thanks for the suggestion of applying this technique in other tasks. Prior work [1,2,3,4] applied sparsemax successfully to NLP tasks such as machine translation, entailment, and summarization. In this paper we focus on visual attention and on extending fusedmax to 2D, yielding TVmax. We are currently experimenting TVmax in other vision tasks. We believe that our paper may open the door to future application in other tasks.
>
>
> [1] C. Malaviya, P. Ferreira, and A. Martins. Sparse and constrained attention for neural machine translation. ACL 2018. https://www.aclweb.org/anthology/P18-2059.pdf
>
> [2] A. Martins and R. Astudillo. From softmax to sparsemax: A sparse model of attention and multi-label classification. ICML 2016. http://proceedings.mlr.press/v48/martins16.pdf
>
> [3] V. Niculae and M. Blondel. A regularized framework for sparse and structured neural attention. NeurIPS 2017.
> https://papers.nips.cc/paper/6926-a-regularized-framework-for-sparse-and-structured-neural-attention.pdf
>
> [4] B. Peters, V. Niculae, and A. Martins. Sparse sequence-to-sequence models. ACL 2019.
> https://www.aclweb.org/anthology/P19-1146.pdf

---

### Decision · Program_Chairs · 2019-12-19

**Decision:**

Reject

**Comment:**

This paper presents sparse attention mechanisms for image captioning. In addition to recent sparsemax based method, authors proposed to extend it by incorporating structural constraints in 2D images, which is called TVMAX. The proposed methods are shown to improve the quality of captioning, particularly in terms of fewer erroneous repetitions, and obtain better human evaluation scores.
Through reviewer discussion, one reviewer updated the score to rejection. A major concern raised by the reviewers is that the motivation of introducing sparse attention is not clear, and the reason why it improves the quality (particularly, why it can reduce repetition) is not convincing. While we understand it is plausible for long sequences as in text domain, we are not convinced that it is really necessary for image captioning problems. Although authors seem to have some ideas, we cannot see how they will be reflected in the paper so I’d like to recommend rejection.
I recommend authors to polish the paper with a clearer description of the motivation and high-level analysis of the method as well as testing on other visual tasks to show its generality.